# Feature Selection for High-Dimensional and Imbalanced Biomedical Data Based on Robust Correlation Based Redundancy and Binary Grasshopper Optimization Algorithm

**DOI:** 10.3390/genes11070717

**Published:** 2020-06-27

**Authors:** Garba Abdulrauf Sharifai, Zurinahni Zainol

**Affiliations:** 1Department of Computer Sciences, Yusuf Maitama Sule University, 700222 Kofar Nassarawa, Kano, Nigeria; 2School of Computer Sciences, Universiti Sains Malaysia, 11800 Gelugor, Malaysia; zuri@usm.my

**Keywords:** multi-filter, high dimensionality, class-imbalanced dataset, Grasshopper optimisation algorithm

## Abstract

The training machine learning algorithm from an imbalanced data set is an inherently challenging task. It becomes more demanding with limited samples but with a massive number of features (high dimensionality). The high dimensional and imbalanced data set has posed severe challenges in many real-world applications, such as biomedical data sets. Numerous researchers investigated either imbalanced class or high dimensional data sets and came up with various methods. Nonetheless, few approaches reported in the literature have addressed the intersection of the high dimensional and imbalanced class problem due to their complicated interactions. Lately, feature selection has become a well-known technique that has been used to overcome this problem by selecting discriminative features that represent minority and majority class. This paper proposes a new method called Robust Correlation Based Redundancy and Binary Grasshopper Optimization Algorithm (rCBR-BGOA); rCBR-BGOA has employed an ensemble of multi-filters coupled with the Correlation-Based Redundancy method to select optimal feature subsets. A binary Grasshopper optimisation algorithm (BGOA) is used to construct the feature selection process as an optimisation problem to select the best (near-optimal) combination of features from the majority and minority class. The obtained results, supported by the proper statistical analysis, indicate that rCBR-BGOA can improve the classification performance for high dimensional and imbalanced datasets in terms of G-mean and the Area Under the Curve (AUC) performance metrics.

## 1. Introduction

Complex diseases such as brain cancer pose a severe threat to human life. The evolution of microarray technology and the advancement of machine learning, artificial intelligence, and statistical methods have offered new possibilities for the classification and diagnosis of deadliest diseases such as cancer, Alzheimer’s, diabetes etc. The infamous characteristics of microarray datasets are a huge number of features, limited samples, and imbalanced class distribution [1]. Imbalanced class distribution occurs when at least one class is insufficiently represented and overwhelmed by other classes. The training classification model on imbalanced data causes many obstacles to learning algorithms and presents numerous ramifications to real-world applications [2]. This problem causes underestimation of the minority class examples and produces bias and inaccurate classification results toward the majority class examples [1]. Classification of an imbalanced data set becomes more severe with limited number samples and a huge number of features [3,4].

Learning from imbalanced data set has recently drawn interest from the machine learning and data mining communities from both academia and industry, which is reflected in the setting up of various workshops and special issues such as ICML’03 [5], LPCICD’17 [6], and ECML/PKDD 2018 [7]. Various techniques have been proposed to overcome the class imbalance problem, including resampling techniques [8,9,10,11], ensemble learning techniques [12,13,14,15], cost-sensitive learning [16,17,18], one class learning [19,20,21], and active learning [22,23,24]. In resampling techniques, the most widely used methods for the class imbalanced problem are (i) random oversampling (ROS) and (ii) random undersampling (RUS) [25,26]. In the former, random replicates examples from minority class are generated to the original training set, which in some scenarios increases the classification model training time, especially when dealing with high-dimensional data set. In the latter, the examples from the majority class are randomly discarded in order to rectify the disparities between classes. The limitation of RUS is that it discards informative instances that could be useful for the predictive model [25,27]. 

One example of the widely used ROS method is the Synthetic Minority Over-Sampling Technique (SMOTE) proposed by Chawla et al. [8]. This method generates artificial examples for the minority class by interpolating among the neighboring examples of the minority class. SMOTE increases the number of minority class examples by adding new minority class instances from the nearest neighboring instances, which results in improving the classifier generalization capability [28]. Unfortunately, generating artificial instances may not always be the best approach to deal with imbalanced data, especially for sensitive application domains such as biomedical data sets that deal with real data for diagnosis [29]. In this scenario, artificial data might unfavorably affect the classification performance of the diagnosis process. In view of that, techniques that do not seek to modify the current training set in the learning process remain favorable. Recent studies [1,30,31,32] validated the claim that the performance of the imbalanced class methods significantly decline if it is implemented on an imbalanced data set with a huge number of features (the high dimensional data) and a limited number of samples [33]. 

Recently, dealing with imbalanced data sets using feature selection has become popular among data mining and machine learning communities [4,34]. The techniques mentioned earlier (i.e., resampling, etc.) focus on the sampling of training data to overcome imbalanced class distribution [4]. A feature reduction method such as feature selection takes a different approach to overcome the imbalanced class problem instead of over/undersampling the training samples. The general concept is to obtain a subset of features that optimally rectify the disparity among classes in the data set and select the best features that represent both classes. Feature selection approaches are classified into filter methods, wrapper methods, and embedded methods. Filter approaches are computationally efficient to select feature subsets. Filter methods are highly susceptible to being trapped in a local optimum feature subset because their performance is heavily affected by the “feature interaction problem” because the selected features may not be optimal for a specific learning model [35,36]. While wrapper [37,38,39,40] and embedded approaches [41] were presented to select a discriminative feature subset, these techniques can be based on selecting features, where the evaluators are often a cost function, i.e., the contribution of a feature to the performance of the classifiers [8], or the discriminative capability of features [37,38,39]. Selected features in using a loss function may not always yield an optimal performance for the classifier, but ranking features using multi-filter methods and aggregating the outcomes of many filter methods might select discriminative features that achieve better (near-optimal) features that represent minority and majority features, retaining the most informative features that guide the population-based algorithm for the optimal features. In this paper, we examined the imbalanced class problem by considering data sets with a high number of features (high dimensional data) but with small samples [4,26]. 

In this paper, a hybrid filter/wrapper feature selection for a high dimensional and imbalanced class method is proposed based on improved correlation-based redundancy (CBR) and binary grasshopper optimization algorithm (BGOA) as a global population-based algorithm that finds an optimal solution based on fitness combination of highly ranked individual features. Hence the selected subset will be more robust and relevant to the classifier. rCBR-BGOA uses the filter approach (i.e., improved CBR), which initially works to find highly discriminative features. These features are then used by BGOA as a strong initial stage to find the most informative subset of features. The performance of filtering CBR method is improved via ensemble technique in which several filter-based approaches are combined (i.e., ReliefF, Chi-square, Fisher score) to obtain a robust feature list. The top N genes with the highest rank of each subset are merged to form a new data set. CBR is used to improve the filtering stage outcomes. rCBR-BGOA differs from the method in reference [4], which uses a single filter method that is highly susceptible to being trapped in a local optimum and may obtain redundant features, which might increase the complexity of the wrapper process and reduces the performance of the model. Similarly, this approach is more cost-effective than the iterative method [40] that checks for all possible combinations of features. As seen in the subsequent section of experiment results, this means that rCBR-BGOA is a robust and effective method.

rCBR-BGOA addresses high dimensionality and imbalance class well. It is an appropriate method, especially when there is a huge number of features and a need to find the best (near-optimal) proportion of selected positive and negative features. rCBR-BGOA uses a Grasshopper optimisation algorithm to guide the search process more efficiently.

Recently, considerable progress has been made in the classification of a high dimensional and imbalanced data set. However, most of the existing methods deal with only one problem at a time—either imbalanced class distribution or high dimensional data set. In this paper, we only focus on the relevant works in the literature that utilised feature selection approaches to combat the imbalanced data on high dimensional datasets.

Yin et al. [42] overcame the imbalanced class problem using Bayesian learning. This method is based on the expectation that the samples in the majority class significantly influences the overall feature selection process. The proposed method first decomposed the majority class examples into smaller pseudo-subclass. After that, feature reduction approaches were applied to the decomposed examples, where the pseudo-subclasses overcame the imbalanced class distribution across classes. The proposed method counterbalances the impact of the larger class samples on feature selection methods. The experimental results over synthetic features proved that the proposed method is effective in dealing with a high dimensional and imbalanced class data problem. Alibeigi et al. [43] presented a new approach to deal with the high dimensional and imbalanced data using a Density-Based Feature Selection (DBFS), where the attributes are weighed based on their approximated probability density values. This approach begins with assessing the contribution of each attribute, and the highest weighted attributes are selected. The experimental results proved that DBFS is an effective method to overcome feature selection and imbalanced data problem in comparison with other state-of-the-art algorithms. Maldonado et al. [26] overcame the challenge of skewed data and high dimensional data set in the context of binary-classes. This method used the sequential backward selection approach using support vector machine (SVM) and SMOTE using the following three loss functions: (i) balanced loss, (ii) predefined loss, and (iii) standard loss. The proposed methods were evaluated on six imbalanced data sets and recorded better predictive performance in comparison with other state-of-the-art methods.

Zhang et al. [44] introduced a new feature selection and skewed dataset using F-measure instead of classification accuracy as a performance criterion for feature selection method. The structural support vector machine (SSVM) algorithm was based on the maximum F-measure metric to select the relevant features using the weighted vector of SSVM based on the imbalanced data setting. A new feature ranking strategy was proposed, which combined a weighted vector of SSVM and symmetric uncertainty to retain the top-ranking features. Thereafter, a harmony search algorithm was employed to choose the optimal feature subsets. The feature subsets represent the minority and majority classes. The experimental results on six data sets proved that this method is effective in resolving the imbalanced classification of a high dimensional data set. Moayedikia et al. [4] proposed a hybrid technique using symmetric uncertainty and harmony search (SYMON) to deal with the problem of imbalanced class distribution and high dimensional data sets. SYMON employed the symmetric uncertainty to measure the weight of features based on their dependency on classes. The harmony search optimisation algorithm was used to optimises the feature subsets. The proposed method was evaluated in comparison with various similar methods and proved to be effective in dealing with high dimensional and imbalanced datasets.

Viegas et al. [45] developed a new approach to overcome the high dimensional and imbalanced data sets using a genetic programming algorithm. The proposed method combines the most relevant feature sets selected by distinct feature selection metrics to acquire the most discriminative features that improve the predictive performance of the model. This method is evaluated based on biological and textual data sets. The experimental results proved that the proposed method was effective in selecting a small number of most relevant feature subset. Yang et al. [39] proposed an ensemble wrapper method for feature selection on highly skewed data sets. The proposed method retained the highest classification performance of the wrapper-based feature selection approach by simultaneously maximizing the model performance and reducing the selection bias. This method works by training multi-based classifiers with balanced features. Hualong et al. [46] introduced an ensemble method to deal with a multi-class imbalanced classification problem. The authors used one-against all (OVA) coding strategy to convert multi-class into numerous binary classes, each of them performing feature subspace that generates multiple training subsets. Two strategies were introduced—(i) decision threshold adjustment and (ii) random undersampling into each training data to overcome the imbalanced class problem. The proposed method has been evaluated on eight high dimensional and imbalanced data sets, and the experimental results show the proposed method is effective to deal with a multi-class imbalanced data problem. Zhen et al. [47] proposed a novel method, namely WELM, to tackle multi-class imbalance problems both at data and algorithmic levels. At the data level, a class-oriented feature selection method is applied to select features that are highly correlated with minority class samples. At the algorithmic level, extreme learning machine (ELM) was modified to improve the input nodes with high discrimination power, and an ensemble technique is trained to improve the performance of the model. The experiments result on eight gene datasets indicate that WELM is effective and outperforms other methods.

The effectiveness of the proposed method is evaluated on high-dimensional and imbalanced biomedical data sets to evaluate the efficacy of the proposed rCBR-BGOA method. The characteristics of these datasets are different in terms of a number of features, number samples, and class imbalanced ration. The performance of rCBR is evaluated against other filtering methods. Then, BGOA is evaluated by investigating the convergence behaviour of the BGOA using a different combination of BGOA parameters and the number of population sizes. The results of the proposed rCBR-BGOA were compared with state-of-the-art methods using the same datasets in terms of various performance measures, include G-Mean (GM) and Area Under Curve (AUC). The comparative results show that the rCBR-BGOA method is effective and competitive compared to other methods.

## 2. Materials and Methods

### 2.1. Filtering Methods

#### 2.1.1. ReliefF

Relief is a feature weight-based algorithm. It can deal with both continuous and discrete attributes, but the original Relief algorithm can only handle bi-class problems. ReliefF [48] is an improved version of the Relief [49] and is regarded as a distance-based measure [50]. This algorithm operates by randomly picking instances and then finds their nearest neighbour from the same class and other classes. After that, the values of features of these samples are compared to the randomly selected samples and then used in updating the rank of each feature. It progressively picks observations and gives higher ranks to the features that have the same value in the same classes but are different in the opposite class. Therefore, the features are not directly selected, but their weights are gradually updated.

#### 2.1.2. Chi-Square

Chi-square assesses the usefulness of features by measuring the value of the chi-square statistic concerning its relevance with classes [51]. The Chi-Square computation works only with discrete variables. Thus, variables should be discretized from continuous variables to discrete variables. Chi-square (X2) analysis compares the obtained values of the frequencies of the classes in the adjacent intervals that are the same as the N examples. Let $N_{ij}\;$be the number of examples of the$\;C_{i}$ class within the jth intervals, and $M_{ij}$ the number of examples in the jth [52], the expected frequency of $N_{ij}\;$is
(1)$$E_{ij}=\frac{M_{ij}*\left|C_{i}\right|}{N}.$$

Chi-squared is computed using the formula below:(2)$$X^{2}=\sum_{i=1}^{c}\sum_{j=1}^{I}\frac{\left(N_{ij}-E_{ij}\right)}{E_{ij}},$$
where “*I*” represents the number of intervals.

#### 2.1.3. Fisher Score

Fisher score is a commonly used metric. It was proposed by Gu et al. [53]. This algorithm uses a heuristic approach to calculate the independent score for each attribute using the notation of the fisher ratio. Where the $\mu_{f_{i}}$ represent the mean and $\sigma_{f_{i\;}}^{k}$ the standard deviation of the kth class and ith feature respectively. The Fisher score for ith feature $\left(f_{i}\right)$ can be measured as follows:(3)$$F\left(f_{i}\right)=\frac{\sum_{k=1}^{c}n_{k}{\left(\mu_{f_{i\;}}^{k}-\mu_{f_{i}}\right)}^{2}}{\sum_{k=1}^{c}n_{k}{\left(\sigma_{f_{i\;}}^{k}\right)}^{2}},$$
where $n_{k}\;$is the number of samples in class k. After evaluating the Fisher score of each feature, the top-ranked features with the highest scores will be selected. Since the score is evaluated individually, the features selected by the Fisher score is suboptimal. Therefore, it cannot handle redundant features.

### 2.2. Grasshopper Optimisation Algorithm (GOA)

The Grasshopper optimisation algorithm (GOA) was introduced by S. Mirjalili in 2017 [54]. It can be used to solve optimisation problems for superior results [50]. The GOA emulates the swarming behaviour of grasshoppers. The GOA mathematical model is used to simulate the swarming behaviour of grasshoppers, as shown in Equation (4).
(4)Xi=Si+Gi+Ai,
where Xi represents the location of the ith grasshopper, Gi is the gravity strength on the ith grasshopper, Si is the social interaction, and Ai represents the wind advection. The social interaction component can be computed as follows:(5)$$S_{i}=\sum_{\begin{array}{c} j=1\\ j\neq i \end{array}}^{N}s\left(d_{ij}\right)\hat{d_{ij}},$$
where$\;d_{ij}$ is the Euclidian distance between the ith and the jth grasshopper, the distance between grasshoppers can be computed as $\;d_{ij}=\;\left|x_{j}-\;x_{i}\right|$ and $\hat{d_{ij}}$ = $\frac{x_{j}-\;x_{i}}{d_{ij}}$ is a unit vector from the ith grasshopper to jth grasshopper.

The s function, which indicates the social power, is computed as follows:(6)$$s\left(r\right)=fe^{\frac{-d}{l}}-e^{-r},$$
where f defines the strength of attraction and l is the attractive length scale.

The gravity power *G* is computed as follows:(7)$$G_{i}=-ℊ{\hat{e}}_{g},$$
where ℊ represents the gravitational constant and ${\hat{e}}_{ℊ}\;$represents the unity vector toward the centre of the earth.

The third component from Equation (4) is A, which represents the wind advection in Equation (8) is computed as follows:(8)$$A_{i}=u\hat{e_{w}},$$
where *u* is a constant drift and $\hat{e_{w}}\;$is a unit vector in the direction of the wind.

When S, G, and A is substituted in Equation (4), the equation can be rearranged as follows:(9)$$X_{i}=\sum_{j=1\;j\neq i}^{N}s\left(\left|X_{j}-X_{i}\right|\right)\frac{X_{j}-X_{i}}{d_{ij}}ℊ{\hat{e}}_{g}+u\hat{e_{w}}\;.$$

Thus, this mathematical model of the GOA cannot be utilised directly to solve optimisation problems because grasshoppers swiftly reach the comfort area and the swarm does converge to a specified point. In order to solve the optimisation problems, the modified Equation (14) is stated as follows:(10)$$X_{i}^{d}=c\left(\;\sum_{\begin{array}{c} j=1\\ j\neq i \end{array}}^{N}c\frac{ub_{d}-lb_{d}}{2\;}s\left(\left|X_{j}^{d}-X_{i}^{d}\right|\right)\frac{x_{j}-x_{i}}{d_{ij}}\right)+\hat{T_{d}},$$
where $ub_{d}$ and $lb_{d}\;$represents the upper bound and the lower bound in the Dth dimension, respectively. $\hat{T_{d}}$ represents the value of the Dth dimension in the target grasshopper (best solution found so far), and c is a decreasing coefficient that is used to shrink the comfort, attraction and the gravity force is considered to be zero, assuming that the wind direction is towards a target ($T_{d}$) in Equation (10). Parameter c is exerted twice in Equation (10) to control the speed rate of grasshoppers and to balance their exploration and exploitation. The outer c from the left controls the movements of grasshoppers toward the target and balances the exploitation and exploration of the whole population around the target. Meanwhile, the inner c reduces the effect of the comfort zone, attraction zone, and repulsion forces between grasshoppers in order to shrink the space that the grasshoppers should explore and exploit.

#### 2.2.1. Coefficient Parameter

Parameter *c* reduces the comfort zone that is proportional to the number of iterations.
(11)$$c=cmax-l\frac{\mathrm{cMax}-\;\mathrm{cMin}\;}{L},$$
where cMax, cMin represent the maximum and minimum value, respectively, l indicates the current iteration, and *L* is the maximum number of iterations.

Algorithm 1 depicts the pseudo-code of GOA algorithm initial the GOA generates a random initial population and evaluates it using a fitness function after the optimization process starts. After findings the best solution is considered as a target, the algorithm continuously executes the steps until the stopping criterion is met. Firstly, the distance between grasshoppers are normalized in the range of [1,4], secondly, the algorithm updates the position of the search agent using equation (10). Thirdly, the current search agent is prevented from going outside of the search boundaries. Lastly, the algorithm returns the best solution achieved as an approximation of the global optimum.
**Algorithm 1** The main steps of the GOA algorithmInitializes the swarm xi (i=1,2,3,…, n)Initialize cmax, cmin, and maximum number of iterationsCalculate the fitness of each search agentT = the best search agent**while** (I<Max number of iterations)      Update c using Equation (11)      For each search agent       Normalize the distance between grasshopper in [1,4]       Update the position of current search agent by the Equation (10)       Bring the current search agent back if it goes outside the boundaries      End for      update T if a better solution is achieved      solution i=i+1**End while****Return**


#### 2.2.2. Binary GOA

In view of the NP-hard nature of feature selection optimisation, the search space can be used by binary values, thus some of the GOA equations needed to be modified. In GOA, each grasshopper updates its position based on its current position, the position of the best grasshopper found so far (target), and the position of all other grasshoppers as in Equation (10). Mirjalili and Lewis [55] proved that the best approach to convert the optimisation algorithm from continuous to binary without alterations is a significant component of using transfer functions.

The transfer function is used in Equation (12) that is re-defined ΔX in Equation (13), as the probability for changing of the position elements.
(12)$$\Delta X=c\left(\;\overset{N}{\underset{\begin{array}{c} j=1\\ j\neq i \end{array}}{\sum }}c\frac{ub_{d}-lb_{d}}{2\;}s\left(\left|X_{j}^{d}-X_{i}^{d}\right|\right)\frac{x_{j}-x_{i}}{d_{ij}}\right)+\hat{T_{d}}$$

The sigmoidal function is a well-known transfer function proposed in [43] as in Equation (13).
(13)$$\Delta X=\frac{1}{1+e^{-\Delta X_{t}}}$$

The position of the current grasshopper will be updated, as expressed in Equation (13). Based on the probability value ($\Delta X_{t}$) acquired from Equation (13).
(14)$$X_{t+1}^{k}\left(t+1\right)=\{\begin{array}{c} 1\;if\;rand<T\left(\Delta X_{t+1}\right)\;\\ 0\;if\;rand\;\geq \left(\Delta X_{t+1}\right) \end{array}$$

The proposed method for high dimensional and skewed data set has two different phases, namely the multi-filter approach phase and the wrapper approach phase. Each phase will be fully discussed in the subsequent sections.

## 3. The Proposed Method for High Dimensional and Imbalanced Data Using Feature Selection Approach

The proposed approach for high dimensional and imbalanced data set has two different stages, namely, the filter and wrapper approach stage, as shown in Figure 1. In subsequent sections, each stage will be fully discussed.

### 3.1. Stage I: Filter Approach

In this stage, the correlation-based redundancy (CBR) method, CBR formulated using symmetric uncertainty (SU), is modified by coupling its filtering process with an ensemble of three filter methods. Initially, the SU is defined in Section 3.1.1. Then, the modified version of CBR is proposed in Section 3.1.4.

#### 3.1.1. Symmetric Uncertainty

Symmetric uncertainty is the normalized forms of Mutual Information; introduced by Witten and Frank [56]. Normally, correlation is widely used to analyse linear correlation. However, linear correlation cannot measure non-linear dependencies among feature subsets. In view of that, a non-linear correlation measure based on entropy is needed to analyse the non-linear correlation between features, each feature is considered as a random variable (r.v), the uncertainty about the values of random variable A is computed using it is entropy H(A). the entropy of a variable A is computed as follows:(15)$$H\left(A\right)=-\underset{i}{\sum }P\left(x_{i}\right)\log_{2}\left(P\left(x_{i}\right)\right).$$
(16)IG(A|B)≔E(A)−E (A|B)
(17)$$SU\left(A,\;B\right)≔2\;\left(\frac{IG\left(A|B\right)}{E\left(A\right)+E\left(B\right)}\right),$$
where E(A) and E(B) are the entropy of variables A and B, and IG(A|B) is the information gain of A after observing variable B [57,58,59], and SU is the modified version of Information Gain that has a range between 0 and 1.

#### 3.1.2. Combination of Multi-Filter Approaches Phase

Algorithm 2 shows the pseudo-code of the combination of multiple filter methods, among the comprehensive suite of filtering algorithms in the literature, we utilised three of the most commonly used filter methods for feature selection (ReliefF, Fisher score and Chi-square) to select optimal features subsets. The three filter methods could be used to rank genes G to obtain a new gene list from each filter W. G genes in each filter W were sorted in descending order. The top N genes from each filter method were selected to form a new gene list.
**Algorithm 2**1: For each feature in the training set D{ $f_{1}$, $f_{2}$,..., $f_{n}$} and class C2: Initialize parameters, S = ∅, R = ∅,3: k: number of selected filters4: k: selected filters W∈[ ReliefF, Fisher score, Chi−Square]Ensemble of filters5: **for**
i
∈
{1, . . ., k} do6:    **for**
j ∈{1, …., m} do      Employ $W_{i}$ to calculate the statistical scores of each gene $G_{i}$     **End of for**
7: Rank genes based on its scores $G_{i}$ in descending order and get a new gene list $W_{i}G_{j}$, j=1, …., m**End of for**8: produce a new ranking list R by combining k filter methods $W_{i}G_{i}$, i=1, …., k using union operator, in order to consolidate the overlapping genes.


#### 3.1.3. Correlation-Based Redundancy Method

Generally, the features obtained from the output of the multi-filter method G  would involve a considerable number of redundant features, which could reduce the performance of the model or may increase the complexity of the wrapper approach in order to select the optimal feature subsets. Koller et al. [60] introduced a Markov Blanket for the feature selection problem; a feature should be discarded if it contains a Markov Blanket [61]. The redundant feature subsets will be removed with the Markov Blanket. Optimal subsets contain all strongly relevant features, a subset of weakly relevant features that contain only relevant and non-redundant features. Koller et al. [60] used the idea of approximate Markov blanket to obtain non-redundant features using approximate Markov blanket Algorithm 3 shows the pseudo-code of approximate Markov blanket.
**Algorithm 3**1: Initialize parameter S = ∅,2: For each candidate feature $f_{i}$ in S, calculate SU between feature-feature and feature-class to eliminate redundant features using AMb.3: ***For***
 i=1 to$\;n,\;j=i+1,f_{i}\;$first feature$,\;f_{j}\;$second feature4:    calculate$\;SU_{i,c},SU_{i,j}\;$and$\;SU_{j,c}$5:        $if\;SU_{i,c}\geq SU_{j,c}\;$&$\;SU_{i,j}\geq SU_{j,c}\;$then6:                   remove $f_{j}$ i.e., S⟵S−{f}.7:           **else**
8:    Insert $f_{j}$ into output selected features list S⟵S+{f},9:        end10: **End of for**11: **Return:** The optimal subset of non-redundant genes *S*.


#### 3.1.4. Hybrid of Multi-Filter Approaches and Correlation-Based Redundancy Analysis

The main work in this stage is to modify CBR to overcome the weakness induced by single filter-based approaches (as mentioned in Section 1) and thus improve the robustness and stability of CBR. An optimal subset contains strongly relevant features and weakly relevant features that include only relevant and non-redundant features. Therefore, we introduce a novel hybrid of the ensemble multi-filter and Correlation-Based Redundancy approach as follows:
(1)Redundancy: A feature is redundant if it has approximate Markov Blanket ($SU_{i,c}\geq SU_{j,c}\;\&\;SU_{i,j}\geq SU_{j,c}$).(2)Irrelevance: A feature is irrelevant if it is ranking score is below the top N ranking score in all the filter methods.(3)Relevance: A feature is relevant if it is ranking score is amongst the top N ranking-score in at least one filter method.(4)Strong relevance: A feature is strongly relevant if it is the Markov Blanket of the target (class).

We utilised three filter methods (ReliefF, Fisher score, chi-square) in order to select the relevant genes from biomedical data sets and the top-ranking genes in each method is used to create a new ranking list R. The correlation-based redundancy method was employed to obtain non-redundant gene sets from R. Finally, the algorithm returns the optimal feature subsets S. The pseudo code of rCBR algorithm is shown in Algorithm 4.
**Algorithm 4** Hybrid Multi-filter approaches and Correlation-Based RedundancyInput: D$\left\{{\;f}_{1},{\;f}_{2},\;\ldots ,{\;f}_{m}\right\}\;$A dataset with m features, number of filter h, number of union filtered gene n, number of genes subset (S)P, classifier COutput: optimal feature subset Tfor i
∈
{1…, h} do    for j ∈{1, …., m} do          employ $W_{i}$ filter to compute the statistical scores of each gene $G_{i}$    end for   select the top-ranking score in each list $G_{i}$ and get a new gene list $W_{i}G_{j}$, j=1, …, mend forproduce a new ranking list R by aggregating the output k filter methods $W_{i}G_{j}$, i=1, …, k using union the operator.  R/* the union of the list genes */for each candidate feature $f_{i}$ in G, compute the interaction between feature-feature and feature-class, to discard redundant features based on Correlation-Based Redundancy using SU.Initialize S = ∅,***For*** i=1 to n, j=i+1        
$f_{i}\;$first feature, $f_{j}\;$second feature       $calculate\;SU_{i,c},SU_{i,j}\;and\;SU_{j,c}$   ***if***$\;SU_{i,c}\geq SU_{j,c}\;\&\;SU_{i,j}\geq SU_{j,c}\;$then     remove $f_{j}$
$i.e.,\;S⟵S-\left\{f_{j}\right\}$.                else        insert $f_{j}$ into output selected features list $S⟵S+\left\{f_{j}\right\},$         **end if**
**  End of For****Return** optimal feature subsets S


Finally, the kNN classifier is used to evaluate the effectiveness and contribution of each filter method and rCBR using four performance metrics—sensitivity, specificity, G-mean, and AUC. To further explore reduced feature subset and identify a subset of informative features proportion of the minority and majority class, BGOA is used to formulate a feature selection process and seek the better feature subset using the wrapper approach.

### 3.2. Stage II: Wrapper Approach

In this stage, we adapted the BGOA as a wrapper approach used to formulate the feature selection process as an optimisation problem that selects the best combination of features from the positive and negative class. The BGOA, as a global population-based algorithm can handle multi solutions at the same time. The BGOA is used to select the most informative feature subsets from the top N ranked features obtained from rCBR. As a criterion to candidate solution of features, the SVM classifier is used to construct the fitness functions. Our proposed method aims to maximize the predictive performance of the model and select the optimal proportion of the feature from the negative and positive examples. The schematic illustration of the proposed approach for feature selection using high dimensional imbalanced data set is shown in Figure 1. The components and process of the proposed method in Stage II are discussed in the following sections.

#### 3.2.1. Solution Representation

Feature subset selection problems are considered as a combinatorial optimisation problem, in which the search space involves a set of all possible subsets [62]. GOA was adapted as a subset selection method to find the near-optimal feature subset from the selected features acquired from rCBR stage. The feature selection problem is recognized as an NP-hard problem [63], which has a huge combinatorial search space in nature. The possible number of solutions in the search space increases exponentially when the number of feature increases, and there are [2^N^] possible feature subsets, where N denotes the number of features. The entire set of features (i.e., solution x) is indicated by a binary string of length N, x = ($x_{1}$, $x_{2}$, …, $x_{N}$), where a bit $x_{i}$ in the solution is set to 1 if the corresponding feature is retained and set to 0 if it is to be discarded. N is the original number of all genes.

#### 3.2.2. Fitness Function

The traditional feature selection methods do not pay attention to the problem of the imbalanced data set [64]. Their objective functions are formulated using classification accuracy or error classification rate and feature subset length, as shown in Equation (18),
(18)$$\mathrm{Fitness}\;=\alpha \gamma R\left(D\right)+\beta \;\left(\frac{\left|N-R\right|}{\left|N\right|}\right),$$
where γR(D) is the average classification accuracy rate acquired, |R| is the amount of feature selected, and |N| are the original features, α and β are two parameters to reflect the role of classification rate and length of the subset, α ϵ|0, 1| and =(1−
α).

Classification accuracy is a well-known evaluation measure for classification problems [65]. Nonetheless, it is not an appropriate evaluation measure for extreme imbalanced data set; the number of majority class samples outnumbered the number of minority class samples. If we classify all samples in the majority class, the classification accuracy rate is still very high, but the recognition rate of the minority class will be insignificant. This is unacceptable, especially for medical dataset.

G-mean measures the balanced performance of a learning algorithm between two classes [66]. G-mean is an appropriate performance metric for the imbalanced data set. Therefore, the objective function of feature selection can be formulated using the G-mean metric, as shown in Equation (19),
(19)$$\mathrm{Fitness}\;\mathrm{function}\;1=\alpha .G_{mean}+\beta .\left(\frac{\left|N-R\right|}{\left|N\right|}\right),$$
where α,β, N, R are the same in Equation (18) G-mean is the standard and most well-known evaluation criterion used for imbalanced data set classification. It is the root of the product of class-wise sensitivity, as shown in Equation (20).

According to the confusion matrix, the G-mean measure can be defined as follows:(20)$$G_{mean}=\sqrt{\frac{TP}{TP+FN}\times \frac{TN}{TN+FP}},$$
where the True Negative (TN), False Negative (FN) (i.e., FN is the positive sample, but wrongly classified as negative sample), True Positive (PN) and False Positive (FP) (i.e., FP is the negative sample, but wrongly classified as positive sample). Here positive class named as class 1 and a negative class named class 2. From Table 1 confusion matrix, TP + FN is number samples that belonged to class 1 ($size_{1})$ and TN + FP is a number of samples that belonged to class 2 ($size_{2}$). TP is the number of samples that belonged to class 1 and correctly predicted. TN is the number of samples that belonged to class 2 and correctly predicted. It is equivalent to True Positive of class 2 ($TP_{2}).$ Therefore, Equation (20) can be rewritten as follows:(21)$$G_{mean}=\sqrt{\frac{TP}{size_{1}}\times \frac{TN}{size_{2}}}=\sqrt{\frac{TP_{1}}{size_{1}}\times \frac{TP_{2}}{size_{2}}}.$$

Similarly, for the multi-class classification problem, the Equation (21) can be extrapolated as follows:(22)$$G_{mean}={\left(\prod_{i=1}^{k}\frac{TP_{i}}{size_{i}}\right)}^{\frac{1}{k}},$$
where $TP_{i}$ is the number of samples that belonged to class i and were correctly classified to belong to a class i. $size_{i}$ is a number of samples that belonged to class i. In medical dataset classification TP/(TP + FN) is called ‘sensitivity’, and TN/(TN + FP) is called ‘specificity’ [67]. 

AUC is the second performance metric used to formulate the second fitness function. AUC is a numeric value that illustrates the trade-off between the true positive rate (TP) and the false positive rate (FP) in the entire example of classification distribution of a data set [4,39,42]. The AUC is directly computed by sorting the classification probability of each sample, thus computing the trade-off value of the TP and FP at each classification threshold and computing the area under the trade-off values. The AUC evaluates the performance of the classification model discriminative ability between TP and FP without considering misclassification costs.
(23)$$\mathrm{Fitness}\;\mathrm{function}\;2=\alpha .AUC+\beta .\left(\frac{\left|N-R\right|}{\left|N\right|}\right)$$

The AUC for a two-class classification problem can be modified to deal with multi-class imbalance problems. MAUC is defined for the multi-class problem as follows:(24)$$MAUC=\frac{\sum_{i=1}^{m}AUC_{i}}{m},$$
where i is the index of class to be considered.

The AUC can be computed using Equation (25).
(25)$$AUC=\frac{1+TP_{rate}-FP_{rate}}{2}$$

## 4. Experimental Results and Discussion

This section discusses the effectiveness of rCBR-BGOA in dealing with high dimensional and imbalanced data sets. Table 1 reports the description of the nine benchmark microarray data sets used to evaluate the proposed method. The BGOA was utilised to search for optimal feature subsets with the maximum G-mean and AUC using a SVM classifier. The SVM was chosen due to its simplicity and common usage in existing papers. Since most of the data sets have a small number of samples in the training set, the reported results are averaged over five iterations to avoid feature selection bias [68,69]. We utilized MATLAB 2019a [70] to implement our proposed method. The GOA is adapted as a feature subset generation to represent the wrapper approach to figure out the near-optimal feature subset from the reduced set of features generated by the filtering stage. The experiments were performed on an Intel Core i5-4300U CPU @ 1.90 GHz CPU and 8 GB RAM in Microsoft Windows 10 Pro platform personal computer. The maximum number of iterations (L) is set to 100, and the number of search agents (N) is 50. The α and β values in the Equations (21) and (23) are set to 0.9 and 0.1, respectively.

### 4.1. Dataset

To validate the proposed method, we carried out several experiments on thirteen high dimensional skewed biomedical data sets that were obtained from the microarray data set repository [71,72,73], and https://www.gems-system.org/, the characteristics of these data sets are summarised in Table 2. The imbalanced ratio (IR) between positive and negative class labels are shown in Table 2.

### 4.2. Effect of the Proposed Filtering rCBR Method and Contribution of the Individual Filter Method

In order to study the impact of the proposed rCBR on the performance of the classification process, the classification behaviour of rCBR is compared with each basic ReliefF, Fisher score, and Chi-Square worked individually. For comparative purposes, we performed five-fold cross-validation to estimate four performance metrics sensitivity, specificity, G-mean and AUC for all comparative methods.

In terms of G-mean and AUC predictive accuracy, the rCBR obtained the best results for five out of the ten datasets. Additionally, for DLBCL, maximum G-mean predictive accuracy was achieved by rCBR-BGOA and ReliefF, which were higher than the results obtained by the other methods. Finally, rCBR loss for three datasets to other methods, ReliefF obtained best score results for two datasets (i.e., Breast and Brain tumour) and Fisher score recorded the best score results for Lung dataset which were slightly higher than the result acquired by the rCBR method. From Table 3, we can see that the proposed method is able to outperform other filter-based feature selection algorithms in most of the tested datasets. Therefore, the proposed rCBR method achieves promising results in the improvement of the performance of the classifier. These outstanding results show that the ensemble multi-filter method has empowered the robustness and stability of the correlation-based method.

### 4.3. Effect on the Performance of BGOA

As a preliminary evaluation, we experimented with the effect of the population size on the performance of the basic approach, which is BGOA, in terms of G-mean predictive accuracy. Therefore, the BGOA was evaluated at different population sizes (i.e.,10, 20, 30, 40, 50, and 60). Figure 2 shows the impact of changing the population size on the attained G-mean predictive accuracy for all datasets. We observed that the range of variation in the G-mean predictive accuracy in most of the datasets is minimal. In addition, increasing the population size does not always improve the results. Therefore, we set the population size to 50 for all next experiments as a trade-off between the classification accuracy and the overhead of the running time of the algorithm.

To evaluate the proposed method, we compared rCBR-BGOA against similar works in the literature such as Guyon et al. [74], Maldonado et al. [26], Yin et al. [42], Moayedikia et al. [4], and Chunkai et al. [44]. Additionally, to validate the effectiveness of the rCBR-BGOA in comparison with SMOTE [75,76] as a well-known baseline method for imbalanced class algorithms, we utilised the two well-known filtering and ranking algorithms of ReliefF (RLF) and Principal Component Analysis (PCA), in combination with SMOTE. The combinations are called SMOTE-RLF and SMOTE-PCA. We evaluate rCBR-BGOA with high dimensional and imbalanced data set with varying percentages of imbalanced data distribution on nine biomedical data sets to evaluate the proposed method against the state-of-the-art methods. The experimental results are reported in Table 4 and Table 5 for two performance metrics of G-mean and AUC, respectively.

The results are summarised in Table 4 and Table 5. Table 4 reports the G-mean predictive accuracy obtained by the proposed rCBR-BGOA and eight methods. We observed that in terms of G-Mean predictive accuracy, the rCBR-BGOA performed very well on five data sets (i.e., COL, CNS, LUNG, GLIOMA, and DLBCL). The G-Mean predictive accuracy obtained by the rCBR-BGOA were significantly higher in comparison to the G-mean predictive accuracy recorded by other algorithms. In the CAR data set, the feature ranking based on harmony search (FRHS) method achieved a G-mean of 100%, which was slightly higher than the value obtained by rCBR-BGOA. Similarly, in the Leu data set, the SYMON algorithm achieves the highest G-mean predictive accuracy obtained of 100%, which was slightly higher than the value obtained by our algorithm. In the LUG data set, four algorithms tied to achieve the highest G-mean predictive accuracy obtained of 100%. The results showed that rCBR-BGOA is an effective and efficient algorithm to overcome high dimensionality and imbalanced data set. Table 5 reports the AUC predictive accuracy obtained by various algorithms that were used to evaluate our proposed method. rCBR-BGOA achieved the highest AUC predictive accuracy obtained across five data sets (i.e., Colon, CNS, DLBCL, Leu and LUG) which were significantly higher than the AUC predictive accuracy obtained by other algorithms. In the SRBCT, rCBR-BGOA tied with other algorithms to obtain the highest AUC predictive accuracy of 100%. In the CAR and BREAST datasets, the SYMON achieved the highest AUC predictive accuracy obtained, which was slightly higher than the AUC predictive accuracy obtained by our algorithm.

### 4.4. Statistical Analysis

The Wilcoxon signed-rank test was used to evaluate the statistical significance of the differences between the resulting G-mean predictive accuracy of rCBR-BGOA against methods. The purpose of this test is to evaluate if the results from the two methods are independent. The null hypothesis claims no significant difference between the proposed rCBR-BGOA approach and other approaches. When the significance level is greater than 5%, the null hypothesis is retained, implying no significant improvement using the proposed rCBR-BGOA approach. In Table 6, the Wilcoxon signed-rank test was computed based on pairwise algorithm comparisons for G-mean predictive accuracy. The resulting test p-values are less than the significant level of 0.05 for the seven methods. Therefore, there are statistically significant differences between the test G-mean predictive accuracy acquired by the rCBR-BGOA and the G-mean predictive accuracy obtained by the other seven methods. However, the resulting G-mean predictive accuracy of the Wilcoxon test for the FRHS exceeded the significance level of 0.05. Thus, there is no statistical difference between the test G-mean predictive accuracy acquired by the rCBR-BGOA and that of FRHS method. Table 7 reports the resulting test values of the Wilcoxon test for the AUC predictive accuracy for the six methods. There are statistically significant differences between the test AUC predictive accuracy obtained by the rCBR-BGOA and the AUC predictive accuracy obtained by the other five methods. However, the G-mean predictive accuracy of the Wilcoxon test for the SYMON exceeded the significance level of 0.05. Thus, there is no statistical difference between the test G-mean predictive accuracy acquired by the rCBR-BGOA and that of SYMON method.

### 4.5. Execution Time

The performance of the rCBR-BGOA was evaluated based on total execution time in comparison with recent proposed works in the literature. We measured the runtime on the total execution time for both feature ranking time and feature selection time for rCBR-BGOA and then compared with SYMON, SVMRFE, D-HELL, SVM-BFE, SMOTE-RLF, and SMOTE-PCA. The results are reported in Table 8 The overall runtime of rCBR-BGOA is a hybrid of filter ranking and feature selection step using the Grasshopper optimisation algorithm, which considers feature selection as an optimisation problem. rCBR-BGOA execution time was better than a similar evolutionary method, such as SYMON, but failed to compete with other feature-weighing approaches, such as D-HELL, SVM-REF, and SMOTE_PCA.

In practice, numerous classification applications have more than two class labels with imbalanced class distribution, such as the classification of multiple tumours [77,78]. The multi-class imbalance problem poses severe challenges that cannot be observed in the binary classification problem. Wie et al. [79] reported that handling multi-class imbalanced problems with different misclassification costs of classes is more demanding than the binary classification problem. The binary-class algorithms are less effective in dealing with multi-class imbalanced tasks [79,80]. Various coding strategies to deal with multi-class imbalanced problem have been proposed in the literature, such as the one versus all (OVA) [81,82] method, the one versus one (OVO) [83,84] method, error-correcting output codes (ECOC) [85], and decision directed acyclic graph (DDAG) [86]. These strategies have been applied to classify multi-class high dimensional imbalanced data. Statnikov et al. [87] investigate these strategies on high dimensional and imbalanced data. After performing several experiments using various datasets with varying imbalanced ratio, the authors suggested that OVA often achieve better results in comparison with other coding strategies.

In view of the aforementioned problem, the rCBR-BGOA method was extrapolated to deal with multi-class high dimensional and imbalanced biomedical data sets. The proposed method is evaluated and compared with other works in literature that were proposed for similar problems, such as Hualong et al. [46] and Zhen et al. [47]. 

Table 9 reports the results obtained by the proposed rCBR-BGOA method on four datasets and compared with other state-of-the-art methods. The results on Brain Tumor1 data show that the G-mean predictive accuracy obtained by the rCBR-BGOA was 97.9% which was significantly higher than the G-mean predictive accuracy obtained by the other methods.

In the Brain-Tumor2 dataset, rCBR-BGOA achieved the highest G-mean predictive accuracy of 98.8, which was significantly higher than the G-mean predictive accuracy obtained recorded by other algorithms. In the Lung-Cancer dataset, a maximum G-mean predictive accuracy of 97.2% was achieved using Class oriented feature selection and Ensemble Modified WELM (C-E-MWELM), which was slightly higher than the G-mean predictive accuracy obtained by the rCBR-BGOA method. In the SRBCT dataset, the maximum G-mean predictive accuracy of 100% was achieved by the rCBR-BGOA and EnSVM-OAA, which were slightly higher than the G-mean accuracy obtained by the C-E-MWELM method. Therefore, the rCBR-BGOA algorithm that was proposed in this paper could obtain a subset of the best features proportion of minority and majority class with the highest correlation and low redundancy. rCBR-BGOA could effectively improve the predictive performance of the classification algorithm.

## 5. Conclusions

This paper aims to address the problem of high dimensional and imbalanced biomedical data sets using a feature selection approach to select optimal feature subsets that represent the minority and majority class. For this purpose, we introduced rCBR-BGOA as a novel approach to overcome this problem. rCBR-BGOA is a two-phase algorithm. In the first stage, three filter methods are employed to select the relevant features, while the correlation-based redundancy is used to select the proportion of the non-redundant features of both classes. In the second stage, the relevant and non-redundant feature subsets obtained from the filtering phase are used as an input to BGOA. BGOA works as a wrapper approach to select the best (near-optimal) feature subsets proportion of both majority and minority class using two fitness functions of G-mean and AUG in terms of class imbalance setting. These evaluation measures are more precise than classification accuracy measures to reflect the classification performance on high dimensional and imbalanced data sets. The rCBR-BGOA was evaluated in comparison with other state-of-the-art methods. The results obtained demonstrated that rCBR-BGOA was comparable or better than other state-of-the-art methods on the thirteen benchmark data sets used to evaluate the proposed algorithm. In terms of the G-mean predictive accuracy, our proposed rCBR-BGOA method outperforms other proposed methods and state-of-the-art algorithms on eight data sets. Similarly, in terms of the AUC predictive accuracy, the rCBR-BGOA outperforms other competitors on five data sets. The experimental results obtained proved that the rCBR-BGOA method can select the best feature subsets that represent both the majority and minority classes. The proposed rCBR-BGOA method was effective due to the integration of rCBR filter approach and the BGOA wrapper approach, which resulted in selecting a fewer number of salient features from both classes.

On the limitations, the rCBR-BGOA has one that will be addressed for future work. This limitation is its computation time. We observed that the rCBR-BGOA execution time can be improved using an adjustable parameter to control a maximum number of iteration to be used so that the searching through feature search space will terminate as soon as the highest value of the fitness function is attained at any number of generation instead of a fixed number of iterations. In some situations, the rCBR-BGOA can achieve the highest value of the fitness function at a fewer number of iterations. Still, the rCBR-BGOA continues until the maximum number of iterations is satisfied, which significantly increases the computationally expensive of the algorithm. The rCBR-BGOA MATLAB source code is available as Appendix A.

## Figures and Tables

**Figure 1 genes-11-00717-f001:**
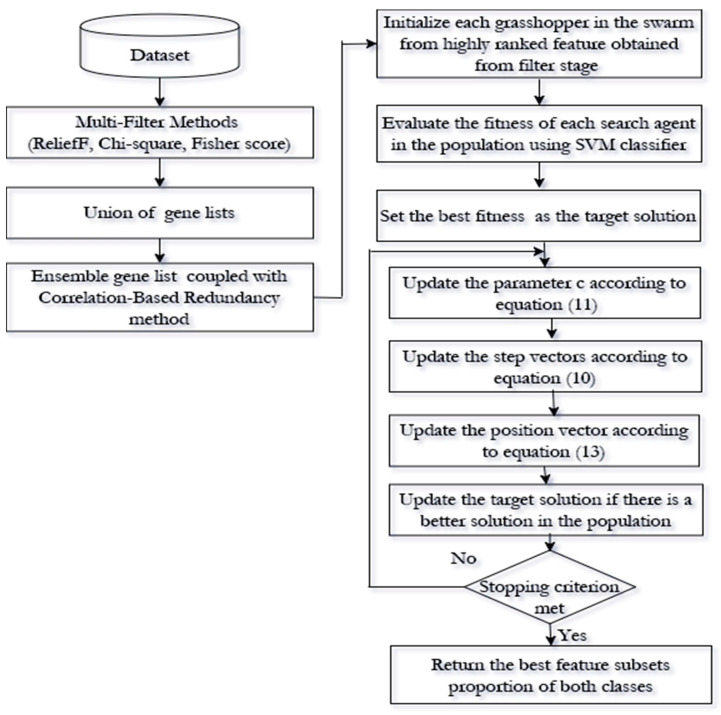
Schematic illustration of the proposed approach for high dimensional imbalanced data set using feature selection.

**Figure 2 genes-11-00717-f002:**
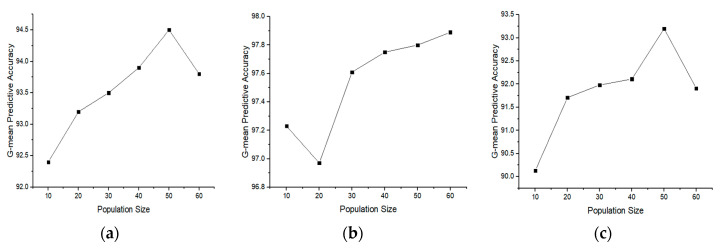
(**a**–**i**) The G-mean accuracy results of rCBR-BGOA with different population sizes for Breast, CAR, CNS, Colon, DLBCL, GLIOMA, Lung, Leukemia, and SRBCT datasets. (**a**) Breast dataset. (**b**) CAR dataset. (**c**) CNS dataset. (**d**) Colon dataset. (**e**) DLBCL dataset. (**f**) GLIOMA dataset. (**g**) Lung dataset. (**h**) Leukemia dataset. (**i**) SRBCT dataset.

**Table 1 genes-11-00717-t001:** Confusion matrix.

	Predicted Positive Class 1	Predictive Negative Class 2
Actual positive class	TP (True Positive)	FN (False Negative)
Actual negative class	FP (False Positive)	TN (True Negative)

**Table 2 genes-11-00717-t002:** Data sets and their characteristics used in this paper for evaluation.

Data Sets	#Features	#Samples	#Classes	IR
Colon	2000	62	2	1.82
DLBCL	7129	59	2	1.04
CNS	7129	60	2	1.86
Leukaemia	7129	72	2	1.88
Breast	24482	97	2	1.11
CAR	9182	174	2	14.82
LUNG	12534	181	2	4.84
GLIOMA	4433	50	2	2.43
SRBCT	2308	83	2	6.55
Brain_Tumor1	5920	90	5	15.00
Brain_Tumor2	10367	50	4	2.14
SRBCT_4	2308	83	4	2.64
LUNG Cancer	12601	203	5	23.17

**Table 3 genes-11-00717-t003:** Comparison of the proposed rCBR method against single filter-based methods in terms of G-mean, AUC, Sensitivity (Sens) and specificity (Spec) performance measures.

Dataset	Measures	ReliefF	Fisher Score	Chi Square	rCBR
Breast	G-mean	66.9	**67.2**	61.3	66.7
	AUC	66.7	67.3	61.5	**67.4**
	Sens	61.6	68.1	59.7	65.2
	Spec	71.8	66.4	63.4	69.3
Colon	G-mean	80.5	82.1	68.1	83.9
	AUC	81.1	82.9	69.4	84.9
	Sens	80.4	85.1	70.9	97.1
	Spec	81.6	80.7	68.0	72.6
	AUC	30.8	73.2	71.5	83.9
CAR	G-mean	30.7	67.7	50.9	73.5
	Sens	30.7	47.1	46.6	70.0
	Spec	95.9	99.4	96.4	97.6
CNS	AUC	77.1	74.2	66.4	85.6
	G-mean	76.4	71.1	65.6	85.1
	Sens	78.3	55.1	54.0	83.8
	Spec	75.9	93.3	79.7	88.1
DLBCL	AUC	98.0	96.7	86.0	98.0
	G-mean	97.9	96.5	85.1	97.9
	Sens	96.0	93.3	83.4	96.0
	Spec	100	100	66.7	100
	AUC	89.7	97.2	87.6	99.8
Leukemia	G-mean	89.1	97.0	86.9	99.0
	Sens	100	100	77.6	100
	Spec	79.6	94.4	97.5	98.8
Lung	AUC	97.8	99.0	94.9	97.3
	G-mean	97.8	99.0	94.7	97.2
	Sens	95.6	98.0	89.8	94.5
	Spec	100	100	100	100
Glioma	AUC	30.8	81.7	30.4	95.0
	G-mean	30.9	81.7	30.9	95.0
	Sens	30.7	63.3	30.7	90.0
	Spec	89.3	100	86.1	100
SRBCT	AUC	98.7	96.7	90.8	100
	G-mean	98.6	96.3	89.9	100
	Sens	100	93.3	81.7	100
	Spec	97.4	100	100	100
Brain Tumour1	AUC	89.6	85.9	74.00	86.2
	G-mean	88.9	85.1	72.9	86.2
	Sens	100	83.3	70.0	90.0
	Spec	79.1	88.5	78.0	82.4

**Table 4 genes-11-00717-t004:** Experimental results of rCBR-BGOA in comparison other methods on G-Mean metric. F/5, 2F/5, and 3F/5 are 20%, 40%, and 60% of the total features in F.

Data Sets	d	rCBR-BGOA	SYMON	SSVM-FS	FRHS	SVM-RFE	SVM-BFE	D-HELL	SMOTE-RLF	SMOTE-PCA
BREAST	F/5	92.8	62.6	-	-	56.0	66.4	62.6	52.4	52.4
	2F/5	91.8	62.6	-	-	62.6	58.0	62.6	58.5	52.4
	3F/5	90.5	62.6	-	-	62.6	52.0	66.4	52.4	52.4
CAR	F/5	97.1	93.5	96.4	100	90.5	95.3	88.7	98.5	90.5
	2F/5	97.4	93.5	98.2	100	92.4	93.6	90.5	95.3	90.5
	3F/5	97.4	93.5	98.2	96.5	92.4	92.4	88.7	95.3	90.5
CNS	F/5	95.2	79.0	-	-	74.5	74.5	70.7	57.7	62.4
	2F/5	94.1	79.0	-	-	74.5	69.7	74.5	66.7	74.5
	3F/5	94.2	79.0	-	-	74.5	74.5	74.5	74.5	74.5
Colon	F/5	94.6	67.4	78.2	74.6	60.0	56.0	67.0	60.3	58.5
	2F/5	93.1	71.5	71.4	74.6	60.0	60.0	60.0	60.3	63.0
	3F/5	93.1	67.4	71.4	74.6	56.0	56.0	64.0	60.3	63.0
DLBCL	F/5	100	29.6	54.3	76.2	25.0	25.0	22.3	27.4	38.7
	2F/5	100	29.6	58.8	76.2	27.3	25.0	25.0	27.4	54.7
	3F/5	100	29.6	62.6	78.4	27.3	25.0	25.0	2.4	29.6
Leukemia	F/5	99.4	100	-	-	31.6	31.6	50.0	0.7	0.7
	2F/5	99.7	100	-	-	31.6	31.6	83.6	0.7	0.7
	3F/5	99.7	100	-	-	31.6	31.6	44.7	0.7	0.7
LUNG	F/5	100	100	-	-	97.3	100	100	96.8	96.8
	2F/5	100	100	-	-	97.3	100	100	96.8	96.8
	3F/5	100	100	-	-	97.3	100	100	96.8	96.8
GLIOMA	F/5	96.6	88.7	85.6	91.5	75.4	92.6	72.6	82.5	84.2
	2F/5	96.5	85.4	83.4	92.8	73.8	90.2	80.6	82.5	84.2
	3F/5	97.3	81.3	79.3	89.6	73.2	86.7	82.6	79.4	84.2

SYMON: symmetrical uncertainty and harmony search, FS: Feature selection; FHRS: feature ranking based on harmony search; RFE: recursive feature elimination; BFE: backward feature elimination; HELL: Hellinger distance; RLF: ReliefF; PCA: Principal component analysis

**Table 5 genes-11-00717-t005:** AUC performance metric on microarray datasets and variant data sizes for the different methods F/5 means 20% of the total features in F.

Data Sets	d	rCBR-BGOA	SYMON	SVM-RFE	SVM-BFE	D-HELL	SMOTE-RLF	SMOTE-PCA
BRE	F/5	78.1(0.2)	79.2(0.8)	75.0(0.2)	29.1(0.4)	75.0(0.4)	65.2(0.8)	59.3(0.2)
CAR	F/5	96.9(0.2)	100(0.2)	75.0(0.2)	29.1(0.4)	75.0(0.4)	93.7(0.2)	93.7(0.20
CNS	F/5	94.1(0.2)	65.2(0.2)	46.0(0.6)	65.2(0.4)	65.2(0.6)	75.2(0.2)	75.0(0.4)
Col	F/5	96.4(0.2)	72.0(0.8)	61.3(0.2)	67.0(0.4)	73.8(0.4)	61.6(0.2)	67.6(0.8)
DLBCL	F/5	100(0.2)	78.7(0.2)	68.7(0.6)	68.7(0.4)	68.7(0.6)	63.4(0.2)	63.7(0.8)
LEU	F/5	99.0(0.2)	93.5(0.2)	87.5(0.2)	87.5(0.2)	87.5(0.2)	87.5(0.2)	87.5(0.2)
LUG	F/5	100(0.2)	96.8(0.2)	96.8(0.2)	96.8(0.2)	96.8(0.2)	96.8(0.2)	96.8(0.2)
SRBCT	F/5	100(0.2)	100(0.2)	100(0.2)	100(0.2)	100(0.2)	100(0.2)	100(0.2)

**Table 6 genes-11-00717-t006:** Wilcoxon signed-rank test for G-mean evaluation metric.

Evaluation Metric	Comparison	Hypothesis	*p*-Value	Significant Difference
G-mean	rCBR-BGOA vs. SYMON	Reject at 5%	2.2689 × 10^−4^ (1)	Yes
	rCBR-BGOA vs. SSVM-FS	Reject at 5%	0.0049 (1)	Yes
	rCBR-BGOA vs. FRHS	Retain at 5%	0.0078 (1)	No
	rCBR-BGOA vs. SVM-RFE	Reject at 5%	1.8162 × 10^−5^ (1)	Yes
	rCBR-BGOA vs. SVM-BFE	Reject at 5%	3.8662 × 10^−5^ (1)	Yes
	rCBR-BGOA vs. D-HELL	Retain at 5%	3.8767 × 10^−5^ (1)	Yes
	rCBR-BGOA vs. SMOTE-ReliefF	Retain at 5%	2.0645 × 10^−5^ (1)	Yes
	rCBR-BGOA vs. SMOTE-PCA	Retain at 5%	1.816 × 10^−5^(1)	Yes

**Table 7 genes-11-00717-t007:** Wilcoxon signed-rank test for AUC evaluation metric.

Evaluation Metric	Comparison	Hypothesis Decision	*p*-Value	Significant Difference
AUC	rCBR-BGOA vs. SYMON	Retain at 5%	0.0781	No
	rCBR-BGOA vs. SVM-RFE	Reject at 5%	0.0156	Yes
	rCBR-BGOA vs. SVM-BFE	Reject at 5%	0.0156	Yes
	rCBR-BGOA vs. D-HELL	Reject at 5%	0.0156	Yes
	rCBR-BGOA vs. SMOTE-ReliefF	Reject at 5%	0.0156	Yes
	rCBR-BGOA vs. SMOTE-PCA	Reject at 5%	0.0156	Yes

**Table 8 genes-11-00717-t008:** The total execution time of rCBR-BGOA in comparison with similar methods across data sets.

Execution Time	Algorithms	Data Sets			
		2K (COL)	7K (DLBCL)	12 K (LUG)	24 K (BC)
Execution time	SVM-REF	2.6	7.58	16.67	28.68
	SVM-BFE	80.64	358.52	25,066.6	48,569.73
	D-HELL	2.51	7.57	12.57	21.58
	SYMON	289	2622	17,023	31,805
	SMOTE-RLF	5.045	11.778	196.05	92.32
	SMOTE-PCA	2.755	11.305	59.466	1134.15
	rCBR-BGOA	12.92	17.39	143.19	76.67

**Table 9 genes-11-00717-t009:** G-mean predictive accuracy and compared with other state-of-the-art methods.

Data Sets	rCBR-BGOA	EnSVM-OAA(RUS)	C-E-MWELM
Brain-Tumor1	97.9	40.3	83.0
Brain-Tumor2	98.8	64.6	92.4
Lung-Cancer	96.9	96.2	97.2
SRBCT	100	100	99.9

OAA(RUS): one against all random undersampling; MWELM: modified weighted extreme learning machine.

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
