# Peer review of "Feature Selection for High-Dimensional and Imbalanced Biomedical Data Based on Robust Correlation Based Redundancy and Binary Grasshopper Optimization Algorithm"

_genes, 2020, doi:10.3390/genes11070717_

Round 1

Reviewer 1 Report

Biological datasets are usually highly dimensional and grouped to classes with imbalanced sizes. Inferring meaningful information and applying data science techniques to such data is challenging. There has been several approaches to tackle this problem, but there has been shortcomings. The authors provide an algorithm that tackles the problem by selecting subset of features that represent the data better and allows better discrimination between classes. The feature selection problem here is tackled as an optimisation problem. In short, the algorithm first ranks features individually, removes redundant features and uses a multi-agent approach to search for optimal combination of features that gives optimal discrimination between classes. There are certain ways that I think the method and presentation of it can be improved to make it more visible compared to other similar approaches.

Major revisions:

418: Please explain why you think the method will not be stuck in local minima and returns optimal solution in all times. The problem is that number of agents are fixed to 10 and number of iterations are fixed to 100. Currently, there is no condition to check whether the algorithm is converged or not. This very well might be the reason that the method is slightly less accurate on some of the datasets. So I think it is mandatory to have a better termination condition rather than fixed number of iterations.

119: Please mention shortcomings of other methods in the introduction as it is currently missing in the introduction apart from the SU method. The way it is being described currently does not make your model stand out from the rest. For example, when you mention SU method assumes independence of features, you can highlight the fact that your algorithm first does individual ranking of features and then finds the optimal solution based on fitness of combination of highly ranked individual features. It is also more cost-effective than methods like sequential backward selection that checks for all possibility of combination of features. Hence your approach provides a trade-off between performance and accuracy. It is useful to highlight the fact that your algorithm is a Multi agent algorithm tackling feature selection as an optimisation problem in the introduction.

Minor revisions:

123: Please mention the full name of BGOA as it is the first occurrence of it in the introduction.

150: “I represent” should be changed to “I” represents.

153: Please name the author before reference 39.

203: Explain more on the Grasshopper optimisation algorithm. The formula was explained well about the forces influencing the Grasshopper movement. However, application of this multi-agent approach to finding optimal solution given a new dataset is not well explained.

295: SU in the filtering approach should be explained properly.

338: Please remove the double space.

377& 378: Please reword definitions of TP, FP, FN, TN and mention what they stand for. It does not have to be relevant to positive class or negative class. You can calculate these values per class regardless of the fact that the class counted as positive or negative class.

416: You did not mention if all the algorithms such as filtering and BGOA are implemented by yourself in Matlab from scratch or you use ready software or packages that facilitates implementation of the algorithm.

421: The accuracy of the method is compared with some of the methods introduced in the introduction but not all. It is good if you provide verbal comparison at least with Viegas's genetic algorithm.

423: You mention SMOTE as a baseline for imbalanced class algorithms. Please explain and reference it in the introduction as well.

Suggestions for improvement

Here, I provide some suggestions to improve the algorithm. I do not expect you to implement all bellow suggestions, but it is good to discuss this in the manuscript as one knows how to improve your algorithm and learn about possibilities of using your algorithm in further application areas.

I understand that number of iterations and agents might have been set low due to having comparable performance to other algorithms. However, the performance can be increased with parallelisation. I think the accuracy should not be sacrificed of performance. To ensure having higher accuracy, you can have X parallel iterations for round 1. And then in round 2 you can restart the search from the best performing solutions the algorithm has found in round 1. You can continue that for several rounds and make sure that you increase the accuracy in each round until the algorithm does not find any better solution or the improvement is bellow a small threshold (epsilon). In this way you can ensure the algorithm converges to optimal solution. It also makes it comparable and competent to Viegas's genetic algorithms that you introduce in the introduction.

2- You spend a lot of effort in weighting the features with your three measures. However, those weights are just used to filter out top x features that are fed to the GOA algorithm. I think you can improve the algorithm if you use the weights in the GOA algorithm. The features with higher weights should be more likely to be visited by the agents. This can be achieved with simple modifications in the formulas of movements of agents.

3- You highlighted the fact that the algorithm works only for two-class classification. I think with simple modifications it can be used for multi-class classification as well as regression models. The G-mean you suggested as an accuracy measure that works for binary class classification only. However, you can calculate G-mean for multi-class as well exactly with the same formula. You can calculate G-measure for per-class accuracy estimation as well. In case of a regression problem the fitness function can be easily modified to a regression model accuracy measure such as RMSE or MAE. You may also need to modify slightly your three filter measures as well.

Reviewer 2 Report

Sharifai and Zainoi present rCBRBGOA, a method to improve machine learning algorithms by optimizing feature selection in high-dimensional and imbalanced datasets. They demonstrate good performance on nine independent data sets.

Minor suggestions:

please make the MatLab code available, ideally as a supplement to the article. Please analyze the false-positive and false-negative results for each algorithm and report if these are the same or different for the various algorithms. Are there any biases? Please run the algorithm using various subsets of filters, report on their relative contribution. Page 1, row 14: "However, the high 14 dimensional and imbalanced class has been thoroughly investigated separately." is difficult to understand, please rewrite Page 1, row 41: "In this paper" please move this to the last paragraph in the introduction Page 2, row 48: "resembling," resampling? Page 2, row 76: "disintegrating," correct word usage? Page 6, row 219-221. Indent the for loop Page 11, row 405. Provide DOIs for the data sets for clarity Page 14, row 456. Specify the p-value ct off in the table description. Why does 5 of 6 comparisons have the same p-value? This needs an explanation. What is the number in the parenthesis (1 or 0)? Page 15, row 487: "discard", incorrect word usage.

Round 2

Author Response

The response is uploaded as a word file.

Reviewer 2 Report

The authors have addressed my concerns and submitted an improved manuscript and I will, therefore, recommend Genes to consider this paper for publication. I think the code can be of use to the community and would recommend that the authors make it, or parts of it, available to the community by publishing it on GitHub or similar. However, I will leave this to the editor to decide if this is a requirement or not.

Author Response

Thank you